# Imaging in Renal Cell Carcinoma Detection

**DOI:** 10.3390/diagnostics14182105

**Published:** 2024-09-23

**Authors:** Dixon Woon, Shane Qin, Abdullah Al-Khanaty, Marlon Perera, Nathan Lawrentschuk

**Affiliations:** 1Department of Urology, Austin Health, Heidelberg, VIC 3084, Australia; 2Department of Surgery, The University of Melbourne, Melbourne, VIC 3010, Australia; 3Division of Cancer Surgery, Peter MacCallum Cancer Centre, Melbourne, VIC 3000, Australia; 4Department of Urology, Royal Melbourne Hospital, Parkville, VIC 3052, Australia

**Keywords:** imaging, ultrasound, computed tomography, magnetic resonance imaging, ^18^F-FDG PET/CT, PSMA PET/CT, ^99m^Tc-Sestamibi SPECT/CT, carbonic anhydrase IX, renal cell carcinoma, oncocytoma

## Abstract

Introduction: Imaging in renal cell carcinoma (RCC) is a constantly evolving landscape. The incidence of RCC has been rising over the years with the improvement in image quality and sensitivity in imaging modalities resulting in “incidentalomas” being detected. We aim to explore the latest advances in imaging for RCC. Methods: A literature search was conducted using Medline and Google Scholar, up to May 2024. For each subsection of the manuscript, a separate search was performed using a combination of the following key terms “renal cell carcinoma”, “renal mass”, “ultrasound”, “computed tomography”, “magnetic resonance imaging”, “18F-Fluorodeoxyglucose PET/CT”, “prostate-specific membrane antigen PET/CT”, “technetium-99m sestamibi SPECT/CT”, “carbonic anhydrase IX”, “girentuximab”, and “radiomics”. Studies that were not in English were excluded. The reference lists of selected manuscripts were checked manually for eligible articles. Results: The main imaging modalities for RCC currently are ultrasound, computed tomography (CT) and magnetic resonance imaging (MRI). Contrast-enhanced US (CEUS) has emerged as an alternative to CT or MRI for the characterisation of renal masses. Furthermore, there has been significant research in molecular imaging in recent years, including FDG PET, PSMA PET/CT, ^99m^Tc-Sestamibi, and anti-carbonic anhydrase IX monoclonal antibodies/peptides. Radiomics and the use of AI in radiology is a growing area of interest. Conclusions: There will be significant change in the field of imaging in RCC as molecular imaging becomes increasingly popular, which reflects a shift in management to a more conservative approach, especially for small renal masses (SRMs). There is the hope that the improvement in imaging will result in less unnecessary invasive surgeries or biopsies being performed for benign or indolent renal lesions.

## 1. Introduction

Renal cell carcinoma (RCC) is a renal cortical adenocarcinoma and makes up 90% of renal cancers. RCC makes up 3% of all cancers worldwide [1]. Up to 60% are incidental findings, especially small renal masses, on imaging and the prevalence has been increasing due to the improvement of imaging modality techniques [2]. The primary imaging modalities for detecting RCC are ultrasound, computed tomography (CT), and magnetic resonance imaging (MRI).

The most common type of RCC is clear cell (ccRCC) (70–80%). Other subtypes include papillary RCC (pRCC) (10–15%) and chromophobe RCC (4–5%) [1]. It is difficult to distinguish RCC from oncocytoma or fat-poor angiomyolipoma on imaging due to similar radiological characteristics. As a result, up to 10–25% of small renal masses (SRMs) are unnecessarily resected and found to be benign [3].

Renal biopsy is currently the only reliable method to discern between RCC and oncocytoma prior to resection, with a diagnostic accuracy of 80–90%. However, 10–20% of cases are nondiagnostic and the biopsy of small lesions can be challenging [4,5]. Furthermore, a recent survey created by Warren et al. in collaboration with the European Association of Urology Young Academic Urologist (EAU YAU) Renal Cancer Working Group and the European Society of Residents in Urology (ESRU) found that 48% of healthcare providers offer renal biopsy in <10% of cases, while only 14% offer it to >50% of patients [6].

Due to the shortcomings of conventional imaging modalities, new imaging techniques have been developed to assess renal tumours, including contrast-enhanced ultrasound (CEUS), dual-energy CT (DECT), 18F-Fluorodeoxyglucose (FDG) PET/CT, prostate-specific membrane antigen positron emission tomography/computed tomography (PSMA PET/CT), technetium-99m sestamibi (^99m^Tc-sestamibi) single photon emission computed tomography (SPECT)/CT, and carbonic anhydrase IX (CAIX) inhibitors. The aim is to avoid unnecessary surgery for benign tumours, thus minimising patient morbidity and the loss of renal function. Ultimately, the hope is to diagnose renal cancer using only imaging, eliminating the need for kidney biopsies, which would make the process less invasive and time-consuming for patients while enhancing diagnostic accuracy and reliability.

## 2. Methods

A literature search was conducted using Medline and Google Scholar, up to May 2024. For each subsection of the manuscript, a separate search was performed using a combination of the following key terms “renal cell carcinoma”, “renal mass”, “ultrasound”, “computed tomography”, “magnetic resonance imaging”, “18F-Fluorodeoxyglucose (FDG) PET/CT”, “prostate-specific membrane antigen PET/CT”, “technetium 99m sestamibi SPECT/CT”, “carbonic anhydrase IX”, “girentuximab”, and “radiomics”. Studies that were not in English were excluded. The reference lists of the selected manuscripts were checked manually for eligible articles.

## 3. New Pathological Classification Updates to Renal Tumours

In 2022, the World Health Organisation (WHO) released a fifth classification of renal tumours, and it had significant changes regarding tumour classification. These included a new category of “other oncocytic tumours” with oncocytoma/chromophobe RCC-like features, the removal of the type 1/2 papillary RCC subcategorization, and the addition of eosinophilic solid and cystic RCC as a new independent tumour class. The WHO/ISUP grading is now recommended for all RCC subtypes. The updated classification also includes a new category of molecularly defined renal tumour subtypes [7]. The Genitourinary Pathology Society (GUPS) have also proposed updated classifications to renal neoplasia by introducing three categories—novel entity (validated by multiple studies), emerging entity (good data from some studies, but requires more validation), and emerging entity (limited data available)—with the aim of reducing the category of “unclassifiable renal carcinomas/tumours.” Novel entities include eosinophilic solid and cystic RCC, RCC with fibromyomatous stroma (previously RCC with leiomyomatous or smooth muscle stroma), and anaplastic lymphoma kinase rearrangement-associated RCC. Emerging entities include eosinophilic vacuolated tumour and thyroid-like follicular RCC. Lastly, the provisional entities consisted of low-grade oncocytic tumours, atrophic kidney-like lesions, and biphasic hyalinizing psammomatous RCC [8].

## 4. Ultrasound

Ultrasound is commonly used for imaging kidney lesions, allowing for the differentiation of cystic structures from solid masses, and is often the initial investigation of choice. It is safe, readily available, and inexpensive to perform. Ultrasound can classify minimally complex renal cysts. However, ultrasound is inferior in characterising renal masses compared to other imaging modalities such as CT and MRI and is operator dependent [9]. Siddaiah et al. demonstrated that ultrasound had a sensitivity and specificity of 73.0% (95% CI 66.9–75.9%) and 89.7% (95% CI 74.2–97.2%) in the detection of indeterminate hyperattenuating renal masses initially seen on CT [10]. Contrast-enhanced ultrasound (CEUS) is a newer technique that has gained traction in recent years, offering the better characterisation of renal tumours compared to traditional greyscale ultrasound by allowing the real-time assessment of the renal vasculature. Initially predominantly used for imaging the liver, its application has expanded to other abdominal viscera, including the kidneys, bearing a similar diagnostic capability to CT and MRI while also having a favourable safety record. The ultrasound contrast agents are neither hepatotoxic or nephrotoxic, with no ionising radiation and a short half-life of 5 min [11]. This can be of great benefit to patients with kidney disease who cannot receive contrast agents.

The renal uses of CEUS include discerning between solid tumours, complex cystic lesions, and pseudotumours; the characterisation of indeterminate renal masses; and tumour ablation evaluation [12]. Bosniak classification is the most widely used cancer risk stratification system for cystic renal masses based on contrast-enhanced CT (CECT) or MRI features. CEUS has not been incorporated into the Bosniak classification, but this has been proposed.

CEUS has shown a comparable diagnostic accuracy to CT and MRI and can also possess a greater sensitivity than CT in detecting wall or septa thickening, and additional septal or solid components when evaluating complex cystic masses [13]. This is particularly useful in patients with contraindications to iodinated CT contrast or gadolinium-based MRI contrast. In addition, it is also highly effective in differentiating renal pseudotumours from malignant tumours, including prominent columns of Bertin, persistent fetal lobulation, dromedary humps, abscesses, and areas of renal parenchyma adjoining cortical scarring with compensatory hypertrophy [12]. Spiesecke et al. found that CEUS had a sensitivity of 89% (95% CI 57–98%) and a specificity of 96% (95% CI 80–99%) for pseudotumours [14].

In a study by Zhao et al., both CEUS and MRI displayed an accurate diagnostic ability in differentiating ccRCC from non-ccRCC, with a sensitivity of 89.7% and 91.9% and a specificity of 77.1% and 68.8%, respectively, making CEUS a viable alternative to conventional imaging [15]. Likewise, Chen et al. compared the accuracy of CEUS against MRI for complex renal cysts, which showed a greater accuracy (84.5% versus 78.9%) and sensitivity (97.2% versus 80.6%), but a lower specificity than MRI (71.4% versus 77.1%) [16]. However, Defortescu et al. found that CEUS had a sensitivity of 100%, a specificity of 97%, and a negative predictive value of 100% (k = 0.95) for complex renal cysts [17].

## 5. Computed Tomography

CT is considered the gold standard modality for the evaluation of renal masses. It has a superior accuracy to ultrasound and offers a similar diagnostic performance to MRI, but is much more readily available with a rapid acquisition time. A CT renal protocol allows for the proper evaluation of renal tumours, involving four phases: precontrast, corticomedullary, nephrogenic, and excretory. Precontrast images detect haemorrhage, calcification and fat, and provide a baseline for comparison in postcontrast images. Corticomedullary images differentiate RCC from the other subtypes and highlight the renal vascular anatomy. The nephrogenic phase characterises hypovascular renal masses, and the excretory phase helps delineate the tumour from the collecting system, aiding in the planning of nephron-sparing surgery [18]. However, many renal masses detected are incidentalomas on CT scans that are not dedicated CT renal protocols.

CT is the primary imaging technique for cystic renal masses using the Bosniak classification, in which certain Hounsfield Unit (HU) ranges determine benignity or require further evaluation. A homogenous renal mass of <20 HU or >70 HU on an unenhanced CT is considered benign and does not require further evaluation. A homogenous mass between −10 and 20 HU without enhancement is a simple renal cyst (Bosniak I). A homogenous mass >70 HU on an unenhanced CT is always a hyperdense Bosniak II cyst [19,20].

CECT displayed a diagnostic accuracy of 81.6% for RCC, with a sensitivity of 90.3% and a specificity of 31.2% in a recent study. In comparison, CEUS was found to have an accuracy of 83.5%, a sensitivity of 92.8%, and a specificity of 52.9% [21]. Furthermore, CT is very accurate in identifying RCC that has spread into the renal vein and inferior vena cava, as well as excluding the invasion of the ipsilateral adrenal gland, but can find it difficult to visualise perinephric or renal sinus fat invasion [22].

In terms of delineating RCC subtypes, clear cell RCC displays strong enhancement in the corticomedullary phase (114 ± 44 HU) due to hypervascularisation and washes out in the nephrographic phase (66 ± 24 HU). Papillary RCC is usually homogenous and hypovascular, with a more subtle contrast enhancement of up to 20 HU in the corticomedullary phase, and up to 25% of papillary RCCs do not display enhancement. This subtle enhancement can be mistaken for pseudoenhancement and requires further evaluation with CEUS or MRI [22]. Chromophobe RCC cannot be differentiated from oncocytoma on CT because they share similar features [23]. Collecting duct carcinoma (CDC) is a rare and aggressive type of RCC, which does not have specific distinguishing radiological features. Typical findings on CT include being located in the renal medulla, weak and heterogenous enhancement, renal sinus involvement, infiltrative growth, renal contour preservation, and a cystic component. Furthermore, Young et al. looked for imaging features on CT to delineate CDC and sarcomatoid RCC from other subtypes and elicited a solid mass with an irregular contour or an infiltrative spread pattern as suspicious [24].

The shortcomings of CT include pseudoenhancement, which is an artificial increase in attenuation within a non-enhancing structure thought to be due to the inadequate correction of beam-hardening [25,26]. Most commonly with small, endophytic lesions adjacent to avidly enhancing renal tissue, this can result in a Bosniak I or II cyst being labelled incorrectly as enhancing [27].

A newer technique, DECT, has the potential to improve the characterisation of renal masses using two photon spectra. DECT can distinguish between low-level enhancing tumours and non-enhancing cysts better; differentiate between solid tumours and hyperdense cysts for incidentally found lesions; and reduce the pseudoenhancement of renal cysts with the reconstruction of virtual monochromatic images [19]. DECT enables the simultaneous acquisition of dual-energy images without a significant increase in the radiation dose, typically acquired at 80 and 140 kVp. By imaging the same object at different energy levels, DECT can reconstruct the anatomical structure of the object (conventional CT) and estimate the composition of the materials within it (spectral CT). Each material exhibits a unique spectral response (variation in absorption coefficient) as a function of energy. Consequently, materials with similar absorption coefficients at a single energy level of the radiology spectrum can be differentiated from each other by measuring at two different energy levels. DECT also possesses a greater post-processing power, consisting of monoenergetic images and optimum contrast images, and reduces the signal from metal artifacts. The post-processing algorithms are able to generate virtual unenhanced (VUE) images by deleting the iodine signal in contrast-enhanced CTs, which minimizes the radiation dose and reduces the need for further scans [28]. However, studies have shown varying results with VUE images. Cao et al. found that VUE images reliably detected enhancement in solid masses, but underestimated attenuation in simple and hyperattenuating cysts [29]. Xiao et al. observed that the classification of renal masses for enhancing mass versus non-enhancing cysts did not change with the use of VUE in comparison with TUE images. The diagnostic performance of VUE images for the detection of urolithiasis was inferior to TUE images (AUC 0.79 versus 0.93, *p* < 0.001) because VUE had a reduced sensitivity for the detection of stones of 3 mm or less in diameter (23% [95% CI 12–40%] versus 88% [95% CI 77–94%, *p* < 0.001]. In terms of radiation, the removal of the TUE acquisition would result in an average radiation dose reduction of 30% [30].

Photon-counting detector CT (PCD-CT) is an emerging CT technology that generates electrical signals from incident X-ray photons. It overcomes many of the limitations of conventional energy-integrated detectors, which other CT scanners are based on, and has several advantages, including a greater spatial resolution, a better iodine signal while still enabling multi-energy imaging, the elimination of electrical noise, and an improved dose efficiency. However, with few PCT-CTs available globally, it is still a new technology undergoing validation studies that are yet to look at its utility in RCC [28,31].

Regarding the systemic staging of renal tumours, CT is the most-used modality for the staging of RCC. CT overdiagnoses lymph node spread because of reactive adenopathy. However, it can also produce false negatives due to the presence of micrometastatic disease. Furthermore, CT possesses a decent accuracy in detecting distant metastases. RCC metastases are commonly found in the liver, lungs, bones, and brain and, less frequently, in the pancreas, thyroid, muscle, and skin. As such, CT chest should be included when a renal mass is detected, when practically feasible. RCC metastases are often hypervascular and arterial phase imaging may help in the detection of metastatic disease burden [9,22].

Furthermore, the R.E.N.A.L Nephrometry Score was developed by Kutikov et al. as a quantitative framework for the anatomy of renal masses and consists of (R)adius, (E)xophytic/endophytic, (N)earness to renal sinus or collecting system, (A)nterior/posterior, and (L)ocation relative to renal poles [32]. Zhang et al. found statistically significant associations between all the components of the R.E.N.A.L. Score and the survival outcomes [33].

## 6. Magnetic Resonance Imaging

MRI is indicated in patients who are allergic to iodinated CT contrast and pregnant patients without renal failure [34]. MRI possesses a superior soft tissue contrast, a reduced sensitivity to calcification, and no pseudoenhancement.

MRI renal protocols include non-contrast T1- and T2-weighted sequences, chemical shift imaging and dynamic contrast enhanced 3D gradient-echo sequences. Multiple dynamic acquisitions are used to obtain corticomedullary, nephrogenic, and excretory phase images.

Diffusion-weighted imaging (DWI) with MRI is another area of significant interest that has been investigated. Using apparent diffusion coefficient (ADC) values, diffusion-weighted MRI has been used to discern between benign and malignant renal masses and RCC subtypes. Serter et al. showed no significant difference in the ADC values (lowest ADC and representative ADC) between benign and malignant renal masses (*p* = 0.721, *p* = 0.255). The mean ADC values for ccRCC were significantly higher compared to non-ccRCC (*p* = 0.005, *p* = 0.002) [35]. Similarly, de Silva et al. assessed DWI and ADC in characterising renal tumours. The average ADC value of oncocytoma was significantly higher than RCC and AML (*p* < 0.001). ccRCC had significantly greater ADC values compared to other RCC subtypes and AML [36].

The apparent diffusion coefficient of the DWI images of MRI can also be utilised as a potential imaging marker for diagnosing the local recurrence of RCC, according to Mytsyk et al. The mean ADC of recurrent RCC highlighted significantly greater numbers in contrast to benign fibrous tissue (1.64 ± 0.15 × 10^−3^ mm^2^/s versus 1.02 ± 0.26 × 10^−3^ mm^2^/s (*p* < 0.001)). For the ROC analysis, using an ADC with a threshold value of 1.28 × 10^−3^ mm^2^/s allowed for the differentiation of local recurrence of RCC from benign postoperative changes of the renal parenchyma, with an AUC value of 0.980 (95%CI 0.945–0.998, *p* < 0.001), a 100% sensitivity, and an 80% specificity [37]. There currently exists a lack of standardisation in imaging protocols and no accurate imaging marker for detecting the recurrence of disease after partial nephrectomy (PN). Mouracade et al. observed 1888 abdominal CTs, 118 abdominal MRIs, 236 abdominal ultrasounds, 472 CT chest scans, and 1770 chest X-rays being ordered for surveillance post-PN over an eight-year period. They found that 86.6% of abdominal recurrences were detected on abdominal CT, 10% on abdominal MRI, and 3.4% on abdominal ultrasound; of the thoracic recurrences, 92.3% were on CT chest and 7.7% on chest X-ray [38].

Recently, a clear cell likelihood score (ccLS) has been developed for the diagnosis of ccRCC with multiparametric MRI. It is classified into five levels that provide the likelihood of a renal mass, representing ccRCC ranging from a ccLS5 score with a positive predictive value (PPV) of 93% or “very likely” to a ccLS1 score with a PPV of 5% or “very unlikely”. The ccLS assesses MRI features such as the enhancement presence and degree; macroscopic fat; T2 signal intensity; corticomedullary-phase mild, moderate, or intense enhancement; microscopic fat; and the ancillary features of segmental enhancement inversion, restriction at DWI, and arterial-to-delayed enhancement ratio [39]. Steinberg et al. validated the diagnostic ability of ccLS in a retrospective study of 434 patients across all tumour sizes and stages and found ccLS to be an independent prognostic factor for identifying ccRCC [40].

Kwon et al. found that the specificity of MRI in diagnosing RCC was greatly superior to CT (68.1% versus 27.7%), with similar sensitivities of 91.8% and 94.5%, respectively. Kim et al. compared CECT and MRI in the detection of small renal masses with sensitivities of 79.7% and 88.1%, respectively. However, the specificities of both modalities were low at 44.4% and 33.3%, respectively [41].

For RCC subtypes, Tang et al., in a retrospective study on CDC, reported that 2/22 patients underwent MRI and found that T1-weighted images showed isointense signal shadows while T2-weighted images displayed variable signals [42]. In contrast, Pickhardt et al. observed that 4/17 CDC patients had MRIs with the tumour parenchymal region exhibiting equal signals on T1-weighted images but lower than normal renal parenchyma on T2-weighted images [41]. MRI has also been used to investigate sarcomatoid RCC, a particularly aggressive subtype of RCC, and was found to have T2 low signal intensity areas (T2LIA) in the majority (72.9%) of patients. The volume of T2LIA was also associated with survival [43].

## 7. ^18^F-FDG PET/CT

The most used radiotracer in nuclear medicine imaging is ^18^F-FDG, a glucose analog. It is widely used in the staging of many malignancies including breast, colorectal, oesophageal, head and neck, thyroid, lung, lymphoma, and melanoma. As such, many studies have examined the utility of FDG PET/CT in RCC. However, its use in the primary imaging of RCC is limited due to FDG being physiologically renally secreted, which interferes with the characterisation of renal masses, and it also has low uptake in ccRCC [44]. A meta-analysis by Wang et al. found the pooled sensitivity and specificity of FDG PET/CT for primary renal tumours was 62% and 88%, respectively [45]. In contrast, FDG PET/CT has a better utility for detecting metastatic disease. Ma et al. performed a meta-analysis that revealed that FDG PET/CT had a pooled sensitivity of 86% and specificity of 88% in detecting metastatic lesions [46].

In terms of RCC subtypes, Tang et al. found that 5/23 patients with CDC received a FDG PET/CT where both primary and metastatic tumours displayed increased FDG uptake. The SUV_max_ of the primary tumour ranged from 7.1 to 22.4 and the SUV_max_ of the metastatic lesions were from 6.7 to 36.6 [42]. Similarly, Hu et al. retrospectively observed FDG PET/CT in 2/6 patients with CDC and found that both primary tumours and distant metastases had SUV_max_ values > 2.5 [47].

## 8. PSMA PET/CT

Prostate-specific membrane antigen is a type II transmembrane glycoprotein encoded by the FOLH1 gene that was initially found expressed highly on prostate cancer cells but is also expressed in the tumour-related neovasculature of several other malignancies including RCC [48]. PSMA PET/CT is currently used in the staging of prostate cancer and the detection of recurrent metastatic disease [49,50]. Recently, interest in exploring the use of PSMA PET/CT in RCC has risen as a better molecular imaging alternative to FDG PET/CT (see Table 1). Given the widespread availability of PSMA, many radiotracers have been developed for it, but the two most commonly used tracers are Gallium-68 (^68^Ga) and fluorine-18 (^18^F)-DCFPyL.

In terms of primary tumour imaging, Tariq et al. assessed an intra-patient dual tracer comparison of FDG and PSMA PET/CT against conventional imaging. They showed concordant avidity in 40% of patients, discordant results favouring PSMA PET/CT in 20%, and FDG PET/CT in 40% of cases for primary tumour diagnosis [51]. A prospective study by Wang et al. compared [^68^Ga]Ga-P16-093 PSMA PET/CT with 2-[^18^F]FDG PET/CT in primary and metastatic ccRCC patients. Immunohistochemistry was used to measure PSMA expression in tumour samples taken from either biopsy or surgery. PSMA PET/CT showed significantly better rates of detection than FDG PET/CT in primary ccRCC patients (86.4% versus 59.1%). This was reflected by the maximum standardised uptake value (SUV_max_) of the primary tumour for PSMA PET/CT and FDG PET/CT, with levels of 15.7 ± 9.0 and 5.1 ± 3.4 (*p* < 0.001), respectively [54]. Likewise, Gasparro et al. showed the mean SUV_max_ for primary RCC was 19.3 (range 5.45–54.2) [55]. Aggarwal performed a prospective study on ^68^Ga-PSMA PET/CT in metastatic RCC and found that PSMA PET/CT had the same diagnostic accuracy to CT for primary or locally recurrent tumours in seven patients. The primary lesions also had a significantly greater median SUV_max_ for PSMA PET/CT compared to FDG PET/CT (16.2 versus 5.5, *p* = 0.002), which was included retrospectively for sub-analysis [56].

Furthermore, Chen et al. designed a prospective study comparing CT, ^68^Ga-PSMA-11 PET/CT, and ^18^F-FDG PET/CT in ccRCC with necrosis, sarcomatoid, or rhabdoid differentiation pathological features. Patients underwent preoperative imaging prior to having a nephrectomy. The histopathological features of the surgical specimens were then evaluated and compared radiologically. ^68^Ga-PSMA-11 PET/CT SUV_max_ showed a sensitivity of 100% and a specificity of 75% for tumour necrosis, and a sensitivity of 100% for adverse pathology with a specificity of 80%. ^68^Ga-PSMA-11 PET/CT demonstrated a superior diagnostic ability in identifying the aggressive pathological features of primary ccRCC [53]. Singhal et al. performed a recent meta-analysis on 11 studies that looked at the diagnostic ability of PSMA PET/CT in RCC and found the pooled sensitivity and specificity in the detection of local disease to be 87.2% (95% CI 77–95%) and 100% (95 CI 92.9–100%), respectively [57].

Additionally, PSMA PET/CT has been used to characterise tumour thrombus (TT) in RCCs with venous extension. RCC with TT has an increased risk of mortality and requires aggressive surgical resection in non-metastatic disease. However, current imaging techniques have displayed varying results. MRI is the gold standard for the characterisation of TT, with an accuracy of 76–88%. Tariq et al. explored the use of PSMA PET/CT for TT and showed that 85.7% of TT was PSMA avid. As a result, PSMA PET/CT has the potential to assist in surgical planning [58].

Regarding systemic imaging and the assessment of metastatic lesions, Tariq et al. showed a concordant uptake in 55% of patients in variable regions (bone, lung, retroperitoneal, and paraaortic lymph nodes). There was negative concordance in 27% of cases and a discordant uptake favouring PSMA PET/CT in 18%. Clinical management was changed in three patients due to PSMA PET/CT [51]. Udovicich et al. evaluated PSMA PET/CT in a retrospective cohort of 61 metastatic RCC patients and found that the sensitivity was 91% (95% CI 77–98%) for both PSMA PET/CT and CT, which differed from other studies. They also analysed a subgroup of 40 patients, comparing PSMA PET/CT against FDG PET/CT with detection rates of 88% and 75%, respectively. The SUV_max_ was 15.2 for PSMA versus 8.0 for FDG PET/CT (*p* = 0.02). Management was changed in 49% of the patients due to PSMA PET/CT [52]. Similarly, Wang et al. found that ^68^Ga-P16-093 PET/CT detected tumours in 95.5% of patients compared to 63.6% for FDG PET/CT in the metastatic ccRCC arm. The SUV_max_ also demonstrated a significantly higher uptake from PSMA PET/CT compared to FDG PET/CT, with levels of 11.0 ± 6.4 versus 4.4 ± 2.7 (*p* < 0.001), respectively [54].

Gasparro et al. showed that PSMA PET/CT detected more metastases than conventional imaging in four patients, and the mean SUV_max_ for metastatic RCC lesions ranged from vascular (11.8) to liver (29.8). They highlighted a strong correlation between the positive PSMA PET/CT measured from SUV_max_ and the intensity of PSMA expression on immunohistochemistry in both ccRCC and chromophobe renal cancer. PSMA PET/CT was positive in 68% of ccRCC patients, with a mean SUV_max_ of 34.1; had a moderate uptake in one papillary cancer (SUV_max_ 10.2 in target lesion of iliac bone) and was negative in the other; and had a weak signal in a chromophobe cancer (SUV_max_ 5.5). There was also a significant impact on altering management [55]. Aggarwal et al. revealed that PSMA PET/CT showed more lesions than CT in 27% (10/37) of metastatic RCC patients. PSMA PET/CT detected 55% (59/107) more bone metastases than CT. Conversely, CT found a greater number of liver lesions (5 versus 24, *p* < 0.001). In the FDG PET/CT subgroup analysis, PSMA PET/CT displayed a superior detection of metastases over FDG PET/CT (312 versus 202, *p* < 0.001), in particular, for lung and bone lesions. In terms of RCC subtypes, PSMA PET/CT found a significantly higher lesion number than FDG PET/CT in ccRCC (277 versus 177, *p* < 0.001). However, FDG/PET had a significantly higher tracer uptake in one eosinophilic variant of an ccRCC patient. Moreover, prognostic factors for survival were observed in patients with the involvement of more than two organ systems, a good International Metastatic RCC Database Consortium risk stratification, a high baseline PSMA-Total volume and Total Lesion-PSMA, and a high number of lesions on CT and PSMA PET/CT [56]. The meta-analysis by Singhal et al. found for the detection of metastatic disease, the pooled sensitivity and specificity was 92% (95% CI 86.2–96%) and 96.9% (95% CI 83.9–99.9%). In particular, the pooled sensitivity in the detection of ccRCC was 94.7% (95% CI 88–98.3%), and 75% (95% CI 35–96.8%) for non-ccRCC [57].

## 9. ^99m^Tc-Sestamibi SPECT/CT

^99m^Tc-sestamibi is a radiopharmaceutical used to assess breast, cardiac, and parathyroid pathology. ^99m^Tc-sestamibi collects in tissues that are high in mitochondria and have low multi-drug resistance pump expression, and is elevated in oncocytomas and hybrid oncyocytic/chromophobe tumours (HOCTs), making it a promising diagnostic tool for renal tumours [59].

There have been several clinical trials in recent years on ^99m^Tc-sestamibi SPECT/CT in the risk assessment and characterisation of renal masses (see Table 2). ^99m^Tc-sestamibi SPECT/CT has the best evidence for primary tumour assessment, but minimal evidence in systemic staging. Asi et al. found all oncocytomas had a high uptake of ^99m^Tc-sestamibi in their study. For predicting benign pathology, ^99m^Tc-sestamibi had a positive predictive value of 60% and a negative predictive value of 91.3% [60]. Parihar et al. retrospectively found that the sensitivity and specificity of ^99m^Tc-sestamibi SPECT/CT for diagnosing a benign lesion was 66.7% and 89.5%, respectively, compared to 10% and 75%, respectively for CT [61]. Sistani et al. had a sensitivity of 100% and a specificity of 96% with sestamibi in the detection of benign or oncocytic lesions versus RCCs [62]. A study by Viswambaram et al. showed ^99m^Tc-sestamibi SPECT/CT to have a sensitivity of 89% (95% CI 77–95%) and a specificity of 73% (95% CI 45–91%) in differentiating between malignant and benign renal lesions. However, the diagnostic accuracy did not improve in comparison to visual interpretation alone [63]. Warren et al. demonstrated in a pilot study to examine the diagnostic accuracy of ^99m^Tc-sestamibi SPECT/CT that the sensitivity and specificity to detect benign versus malignant tumours was 100% (95% CI 54–100%) and 85.7% (95% CI 57–98%), respectively. More notably, the sensitivity and specificity to detect oncocytic or chromophobe tumours from other RCCs was 100% (95% CI 74–100%) and 100% (95% CI 63–100%).[64] In a recent retrospective study, Schober et al. found in patients with negative ^99m^Tc-sestamibi imaging (suggestive of malignancy) that 20% who underwent biopsy or surgery returned oncocytoma on histopathology. The negative predictive value was only 80% [65].

In a recent systematic review, Basile et al. included eight studies looking at ^99m^Tc-sestamibi SPECT/CT in the diagnosis of renal masses. Their group found the sensitivity and specificity for oncocytoma and HOCT versus all other renal lesions to be 89% (95% CI 70–97%) and 89% (95% CI 86–92%), respectively. Notably, the sensitivity and specificity of oncocytoma and HOCT versus ccRCC and pRCC was 89% (95% CI 70–97%) and 98% (95% CI 91–100%), respectively. However, the studies are heterogenous, and four (50%) studies were assessed to have a high risk of bias, specifically in patient selection, pathological reporting, and the flow of patients throughout the study [66]. ^99m^Tc-sestamibi SPECT/CT has been also been shown to be cost effective compared to current management strategies such as the surgical resection of tumours that turn out to being benign.

## 10. Anti-Carbonic Anhydrase IX Monoclonal Antibodies & Peptides

Carbonic anhydrase IX is a cell surface glycoprotein enzyme that acts as a buffer in acidic environments and promotes cell survival in malignant cells in hypoxic conditions. The expression of CAIX is high in ccRCC due to the mutation of the von Hippel-Lindau tumour suppressor gene, which activates the hypoxia-inducible factor-1α (HIF-1α) pathway, even in the absence of hypoxia. CAIX is only physiologically expressed in gastrointestinal epithelia and is highly expressed in numerous cancers including lung, breast, pancreatic, colorectal, cervical, and ccRCC [67]. CAIX is not present in renal tissue and is not expressed in chromophobe RCC, papillary RCC, or oncocytoma. Girentuximab is a monoclonal antibody that binds to CAIX and is radiolabelled as a PET/CT tracer. It is excreted through the hepatobiliary system, making it ideal for assessing renal tumours. There have been a few studies assessing the effectiveness of anti-CAIX antibodies in metastatic RCC (see Table 3).

The Renal Masses: Pivotal Study to DEteCT Clear Cell Renal Cell Carcinoma With Pre-Surgical PET/CT (REDECT) trial is the first study that observed that Iodine-124 (^124^I)-labelled girentuximab PET/CT was able to diagnose ccRCC with a greater accuracy than CECT, with a sensitivity of 86.2% (95% CI 75.3–97.1%) versus 75.5% (95% CI 62.6–88.4%), and a specificity of 85.9% (95% CI 69.4–99.9%) versus 46.8% (95% CI 18.8–74.7%), respectively [68]. However, ^124^I has several shortcomings: its high energy positrons reduce image resolution; the simultaneous emission of high energy photons increases background counts, degrading image contrast and accuracy; and ^124^I-iodotyrosine is rapidly expelled from cells. As such, another radiotracer, ^89^Zirconium (^89^Zr), has been investigated for use with girentuximab, with a feasible half-life and better physical properties [72].

Furthermore, Hekman et al. performed a phase I/II study looking at ^89^Zr-girentuximab PET/CT and assessing indistinct renal masses. Their results found a change in the clinical management in 36% of patients and that 21% avoided repeat biopsies [69]. In the IMaging PAtients for Cancer drug selecTion (IMPACT)-renal cell carcinoma (RCC) study by Verhoeff et al., their group found that 70% of lesions were visualised on [^89^Zr]Zr-DFO-girentuximab-PET/CT, 59% on [^18^F]FDG-PET/CT, and 56% on CT. The addition of [^89^Zr]Zr-DFO-girentuximab-PET/CT to CT increased the detection of ccRCC lesions from 56% (95% CI 50–62%) to 91% (95% CI 87–94%), which was greater than [^18^F]FDG-PET/CT and CT combined, of 84% (95% CI 79–88%) [70].

Recently, a new CAIX-binding peptide DPI-4452 has been developed. Radiolabelling DPI-4452 with gallium-68 ([^68^Ga]Ga-DPI-4452) or lutetium-177 ([^177^Lu]Lu-DPI-4452) can be used for investigating patients with CAIX-expressing tumours. Hofman et al. showed in a first in-human phase 1/2 study that [68Ga]Ga-DPI-4452 had a high tumour uptake in ccRCC, with a mean SUV_max_ of 64.6 at one hour and very high tumour-to-background ratios. It was also rapidly eliminated in the blood and urine without significant toxicity displayed, making it a promising tracer for diagnostic purposes in ccRCC [71].

## 11. Radiomics

Radiomics involves the extraction of quantitative metrics from medical images that are not perceivable to the human eye. With the advancement of artificial intelligence (AI) and machine learning, the interest in radiomics has rapidly increased, particularly in the field of oncology. The process consists of image segmentation of the region or volume of interest from CT, MRI, or PET/CT, image processing, feature extraction, and dimension reduction to exclude non-reproducible data.

Nomograms have been developed using CT-radiomics to predict ccRCC nuclear grades and to distinguish between ccRCC and non-ccRCC, with AUCs over 0.88 in both training and validation sets (see Table 4) [73,74]. CT-radiomics has also been used to differentiate between renal masses. Meng et al. found that radiomics could help discriminate sarcomatoid RCC from ccRCC. Their combined radiomics model (corticomedullary phase (CMP), nephrogenic phase (NP), and subjective CT findings) had the highest AUC value 0.974 (95%CI 0.924–0.992), with an accuracy of 93.75, a sensitivity of 96.55%, and a specificity of 88.89%. In contrast, the corticomedullary phase model was the worst performing, with an AUC of 0.772 (95% CI 0.689–0.841), an accuracy of 78.12%, a sensitivity of 65.52%, and a specificity of 82.83%. Yang et al. used radiomics to distinguish renal oncocytoma from chromophobe RCC developing models with radiomic features from unenhanced phase, corticomedullary phase, and nephrogenic phase CT images with promising results. The best performing radiomic model combined radiomic features with clinical factors, with an AUC of 0.93 (95% CI 0.83–1.00) [75,76]. Furthermore, radiomics has been used to assess recurrence, metastasis, and overall survival in patients with RCC. Deniffel et al. assessed the utility of CT-radiomics in predicting the recurrence risk of RCC post-nephrectomy in a retrospective cohort of 453 patients and found that radiomics improved the risk assessment of recurrent malignant disease [75].

A meta-analysis looked at the ability of radiomics in distinguishing between benign and malignant tumours, with a random effects model that showed a log OR 3.17 (95% CI 2.73–3.62, *p* < 0.0001) with a moderate heterogeneity of 74.6% (95% CI 63.7–82.2%, *p* < 0.001). Despite these promising results, it is advised that they should be tempered with caution, as many of the studies were low quality with an inconsistent use of radiomics, resulting in almost 50% of the studies being excluded. The benefit over accuracy from human assessment remains unclear [78]. Moreover, radiomics has been used in the prediction of survival, treatment response, and disease progression in RCC that was treated with immunotherapy and molecular targeted therapies [79]. Khene et al. used K-nearest neighbor, SVM based on linear kernel, random forest tree, and logistic regression to predict the tumour response to nivolumab in metastatic RCC. There were 60.4% of responders in a cohort of 48 patients. The accuracy ranged between 0.71–0.91, the sensitivity from 0.53–1, the specificity from 0.73–0.83, and the AUC from 0.68–0.92 [80].

## 12. Active Surveillance 

The active surveillance of small renal masses has been identified as a viable management alternative because they tend to grow more slowly. Active surveillance has been identified as a follow-up to the serial imaging of a select group of patients with significant comorbidities, where the risk of death/morbidity from intervention is deemed higher than from the renal tumour. There is currently no standardised protocol for the selection of patients or to assess the risk of progression, nor are there standardised surveillance strategies, such as the type of imaging modality and the follow-up schedule. Nayyar et al. performed a systematic review and found that 67% of studies used various imaging modalities in their follow-up protocols, which could lead to inconsistent growth rates as ultrasound has been shown to overestimate tumour size in relation to MRI and CT [81].

## 13. Conclusions

Imaging in renal cancer is a dynamic landscape where the number of SRMs incidentally diagnosed significantly increased with the rise in abdominal cross-sectional imaging in the past few decades, leading to the overtreatment of benign or indolent renal masses. The interest in molecular imaging has spiked in recent years, with a slew of novel radiotracers being developed in a bid to reduce morbidity and the loss of renal function in these patients. In the future, radiomics and AI have the potential to disrupt imaging even further, which can benefit clinical medicine in many areas, ranging from diagnosis, predicting disease recurrence, response to treatment, and survival benefit, which can help determine and stratify management pathways and improve patient outcomes.

## Figures and Tables

**Table 1 diagnostics-14-02105-t001:** PSMA PET/CT in RCC studies.

Study	Year	Type	Patient No.	Objective	Comparator	Histology	Main Outcomes
Tariq et al. [51]	2021	Retrospective	11	To perform an intra-patient dual tracer comparison of FDG and PSMA PET/CT against conventional imaging.	PSMA & ^18^F-FDG PET/CT vs. CT	ccRCC	Primary: 40% concordant, discordant 20% favouring PSMA, and 40% FDG. Systemic: 55% concordant, 27% no concordance, 18% discordance favouring PSMA. PSMA PET/CT changed management in 27% of cases.
Udovicich et al. [52]	2022	Retrospective	61	To assess the impact of PSMA PET/CT in the management of metastatic RCC.	PSMA-PET/CT vs. CT or FDG PET/CT	ccRCC & non-ccRCC	PSMA PET/CT had a detection rate of 84%. PSMA PET/CT changed management in 49% of patients. The SUV_max_ was 15.2 for PSMA and 8.0 for FDG PET/CT (*p* = 0.02). In a subcohort of 40 patients, detection rate was 88% for PSMA and 77% for FDG PET/CT.
Chen et al. [53]	2023	Prospective	72	To compare ^68^Ga-PSMA-11, ^18^F-FDG PET/CT, and CT in ccRCC with necrosis or sarcomatoid or rhabdoid differentiation.	^68^Ga-PSMA-11 vs. ^18^F-FDG PET/CT vs. CT	ccRCC	^68^Ga-PSMA-11 PET/CT performed better than ^18^F-FDG PET/CT and CT in identifying aggressive pathological features of primary ccRCC. ^68^Ga-PSMA-11 PET/CT SUV_max_ showed a sensitivity of 100% and a specificity of 75% for tumour necrosis, and a sensitivity of 100% for adverse pathology, with a specificity of 80%.
Wang et al. [54]	2023	Prospective	44	To compare the diagnostic value of PSMA and FDG PET/CT in ccRCC.	^68^Ga-P16-093 vs. 2-[^18^F]FDG PET/CT	Primary and metastatic ccRCC	PSMA PET/CT had a much higher tumour detection rate than FDG PET/CT. Primary: the detection rate for PSMA PET/CT was 86.4% and 59.1% for FDG PET/CT. SUV_max_ for PSMA PET/CT and FDG PET/CT were 15.7 ± 9.0 and 5.1 ± 3.4 (*p* < 0.001), respectively. Systemic: the detection rate for PSMA PET/CT was 95.5% and 63.6% for FDG PET/CT. SUV_max_ for PSMA PET/CT and FDG PET/CT were 11.0 ± 6.4 vs. 4.4 ± 2.7 (*p* < 0.001), respectively.
Gasparro et al. [55]	2023	Retrospective	26	To find if PSMA expression in renal cancer in primary tumour or metastatic lesions on immuno-histochemistry (IHC) are associated with PET/CT findings.	^68^Ga-PSMA PET/CT and IHC	ccRCC & non-ccRCC	PSMA PET/CT detected more metastases than CT. Positive PSMA PET/CT is linked with moderate or high PSMA expression on IHC. Strong correlation between positive SUV_max_ to intensity of PSMA expression on IHC in ccRCC and chromophobe renal cancer. IHC PSMA score was concordant in primary tumours and metastases. Median survival was significantly higher in negative PSMA PET/CT compared to a positive scan (48 vs. 24 months, *p* = 0.001). There was significant impact on altering management.
Aggarwal et al. [56]	2024	Prospective	37 Subgroup = 15	To assess ^68^Ga-PSMA PET/CT over conventional imaging in metastatic RCC and prognostic impact on outcome.	^68^Ga-PSMA PET/CT vs. CT (subgroup vs. ^18^F-FDG PET/CT)	ccRCC Eosinophilic variant of ccRCC pRCC CDC	^68^Ga-PSMA PET/CT detected more lesions in total than CT (568 vs. 531, *p* = 0.215) PSMA PET/CT detected more lesions than FDG PET/CT (312 vs. 202, *p* < 0.001) with 113 PSMA + FDG- discordant lesions and 14 PSMA-FDG+ discordant lesions. ^68^Ga-PSMA PET/CT tumour burden estimation using Total Lesion-PSMA and PSMA-Total Volume had a prognostic impact on patient survival.

**Table 2 diagnostics-14-02105-t002:** ^99m^Tc-sestamibi SPECT/CT in renal mass studies.

Study	Year	Type	Patient No.	Histology	Outcome
Asi et al. [60]	2020	Prospective	90	10 Oncocytoma 4 AML 2 chronic sclerosis 1 fibroma 1 hydatid cyst ccRCC pRCC chRCC	^99m^Tc-sestamibi SPECT/CT had a PPV of 60% and NPV of 91.3% for predicting benign pathology.
Parihar et al. [61]	2022	Retrospective	42	Malignant/concerning (ccRCC, pRCC) vs. benign/nonconcerning (oncocytic renal neoplasms, chRCC)	The sensitivity and specificity of ^99m^Tc-sestamibi SPECT/CT for diagnosing a benign lesion was 66.7% and 89.5%, respectively, compared to 10% and 75% for CT, respectively.
Sistani et al. [62]	2020	Prospective	29	Oncocytoma HOCT chRCC ccRCC pRCC Mixed pRCC/chRCC	^99m^Tc-sestamibi SPECT/CT had a sensitivity of 100% and a specificity of 96% in the detection of benign or oncocytic lesions vs. RCC.
Viswambaram et al. [63]	2022	Prospective	74	Oncocytoma AML ccRCC pRCC cRCC SCC	^99m^Tc-sestamibi SPECT/CT has a sensitivity of 89% (95% CI 77–95%) and a specificity of 73% (95% CI 45–91%) in differentiating between malignant and benign renal lesions.
Warren et al. [64]	2022	Prospective	20	Oncocytoma Oncocytic RCC chRCC ccRCC pRCC unknown histology	^99m^Tc-sestamibi SPECT/CT sensitivity and specificity to detect benign vs. malignant tumours was 100% (95% CI 54–100%) and 85.7% (95% CI 57–98%). The sensitivity and specificity to detect oncocytic or chromophobe tumours from other RCCs was 100% (95% CI 74–100%) and 100% (95% CI 63–100%).
Schober et al. [65]	2023	Retrospective	60	Oncocytoma chRCC pRCC ccRCC	The negative ^99m^Tc-sestamibi imaging (suggestive of malignancy) showed that 20% who underwent biopsy or surgery returned oncocytoma on histopathology. The negative predictive value was 80%.

**Table 3 diagnostics-14-02105-t003:** Carbonic anhydrase IX antibodies/peptides in RCC studies.

Study	Year	Patient No.	Type	Comparator	Outcome
Divgi et al. [68]	2012	195	^24^I-girentuximab PET/CT	CECT	^24^I-girentuximab PET/CT diagnosed ccRCC with greater accuracy than CECT with a sensitivity of 86.2% (95% CI 75.3–97.1%) vs. 75.5% (95% CI 62.6–88.4%), and specificity of 85.9% (95% CI 69.4–99.9%) vs. 46.8% (95% CI 18.8–74.7%), respectively.
Hekman et al. [69]	2018	30	^89^Zr-girentuximab PET/CT	Nil	^89^Zr-girentuximab PET/CT resulted in a change in clinical management in 36% of patients and 21% avoided repeat biopsies
Verhoeff et al. [70]	2019	42	[^89^Zr]Zr-DFO-girentuximab-PET/CT	CT, [^18^F]FDG-PET/CT	They found 70% of lesions were visualised on [^89^Zr]Zr-DFO-girentuximab-PET/CT, 59% on [^18^F]FDG-PET/CT, and 56% on CT. The addition of [^89^Zr]Zr-DFO-girentuximab-PET/CT to CT increased the detection of ccRCC lesions from 56% (95% CI 50–62%) to 91% (95% CI 87–94%), which was greater than [^18^F]FDG-PET/CT and CT combined of 84% (95% CI 79–88%).
Hofman et al. [71]	2024	3	[^68^Ga]Ga-DPI-4452	Nil	[^68^Ga]Ga-DPI-4452 showed very high tumour uptake in ccRCC, with a mean SUV_max_ of 64.6 at one hour and very high tumour-to-background ratios.

**Table 4 diagnostics-14-02105-t004:** Radiomics in RCC.

Study	Year	Aim	Result
Meng et al. [76]	2020	To differentiate sarcomatoid RCC from ccRCC	The combined radiomics model (CMP, NP, and subjective CT findings) had the highest AUC value 0.974 (95%CI 0.924–0.992) with an accuracy of 93.75, a sensitivity of 96.55%, and a specificity of 88.89%. The CMP + NP model had a similar AUC of 0.966 (95%CI 0.918–0.990), an accuracy of 93.75%, a sensitivity of 89.66%, and a specificity of 94.95%. The CMP model had the lowest AUC of 0.772 (95%CI 0.689–0.841) with an accuracy of 78.12%, a sensitivity of 65.52%, and a specificity of 82.83%.
Zheng et al. [73]	2021	To predict ccRCC grades and detect between ccRCC and non-ccRCC	AUCs of 0.914 in training set and 0.846 in validation set.
Cheng et al. [74]	2023	To discriminate between ccRCC and non-ccRCC	AUCs of 0.982 in training set and 0.949 in validation set.
Denifel et al. [75]	2023	To predict risk of RCC recurrence post nephrectomy	1 of 4 radiomics features was prognostic for disease-free survival with an adjusted hazard ratio of 0.44 (*p* = 0.02). The combined clinical–radiomic model (C = 0.80) was superior to the clinical model (C = 0.78, *p* < 0.001).
Yang et al. [77]	2024	To distinguish renal oncocytoma from chromophobe RCC	Among single phase models, the NP phase model was highest, with an AUC of 0.76 (95% CI 0.57–0.99). Among two phase models, the CMP/NP model was highest, with an AUC of 0.83 (95% CI 0.66–0.99). The model with unenhanced phase, CMP, and NP had an AUC of 0.84 (95% CI 0.69–0.99). The combined model with clinical factors had the highest AUC of 0.93 (95% CI 0.83–1.00).

## Data Availability

Not applicable.

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
