# Peer review of "Imaging in Renal Cell Carcinoma Detection"

_diagnostics, 2024, doi:10.3390/diagnostics14182105_

Round 1
Reviewer 1 Report
Comments and Suggestions for Authors
1. Review of (some) interest.
2. Similar review in recent literature, well elaborated and illustrated:
Recent advances in imaging techniques of renal masses.
Aggarwal A, Das CJ, Sharma S.World J Radiol. 2022 Jun 28;14(6):137-150. doi: 10.4329/wjr.v14.i6.137.PMID: 35978979
3. Some parts of this review well elaborated, other parts rather superficial.
4. Recent new pathologic/histologic classifications of renal masses not mentioned.
5. Ultrasound mentioned for the detection of renal masses: Sensitivity? Operator dependancy?
6. Would it be recommendable to extend the abdominal CT-examination with a Chest-CT when a renal mass is detected (to avoid an additional administration of CM)?
7. Value (or not) of diffusion Weighted MRI?
8. Section on nuclear medicine extensive: adding up all modalities for all renal masses ?
9. Seeding with biopsies of renal masses?
10. Active surveillance?
11. R.E.N.A.L. morphometry ?
...
Mod Pathol. 2021 Jun;34(6):1167-1184.
doi: 10.1038/s41379-021-00737-6. Epub 2021 Feb 1.
Novel, emerging and provisional renal entities: The Genitourinary Pathology Society (GUPS) update on renal neoplasia
Kiril Trpkov # 1, Sean R Williamson # 2, Anthony J Gill # 3, Adebowale J Adeniran 4, Abbas Agaimy 5, Reza Alaghehbandan 6, Mahul B Amin 7, Pedram Argani 8, Ying-Bei Chen 9, Liang Cheng 10, Jonathan I Epstein 11, John C Cheville 12, Eva Comperat 13, Isabela Werneck da Cunha 14, Jennifer B Gordetsky 15, Sounak Gupta 12, Huiying He 16, Michelle S Hirsch 17, Peter A Humphrey 4, Payal Kapur 18, Fumiyoshi Kojima 19, Jose I Lopez 20, Fiona Maclean 21 22, Cristina Magi-Galluzzi 23, Jesse K McKenney 2, Rohit Mehra 24, Santosh Menon 25, George J Netto 23, Christopher G Przybycin 2, Priya Rao 26, Qiu Rao 27, Victor E Reuter 9, Rola M Saleeb 28, Rajal B Shah 29, Steven C Smith 30, Satish Tickoo 9, Maria S Tretiakova 31, Lawrence True 31, Virginie Verkarre 32, Sara E Wobker 33, Ming Zhou 34, Ondrej Hes # 35
PMID: 33526874 DOI: 10.1038/s41379-021-00737-6
Hum Pathol. 2023 Jun:136:123-143.
doi: 10.1016/j.humpath.2022.08.006. Epub 2022 Sep 6.
The 2022 revision of the World Health Organization classification of tumors of the urinary system and male genital organs: advances and challenges
Sambit K Mohanty 1, Anandi Lobo 2, Liang Cheng 3
PMID: 36084769 DOI: 10.1016/j.humpath.2022.08.006
The role of diffusion-weighted MRI and contrast-enhanced MRI for differentiation between solid renal masses and renal cell carcinoma subtypes.
Serter A, Onur MR, Coban G, Yildiz P, Armagan A, Kocakoc E.Abdom Radiol (NY). 2021 Mar;46(3):1041-1052. doi: 10.1007/s00261-020-02742-w. Epub 2020 Sep 15.PMID: 32930832
The diagnostic utility of diffusion weighted MRI imaging and ADC ratio to distinguish benign from malignant renal masses: sorting the kittens from the tigers.
de Silva S, Lockhart KR, Aslan P, Nash P, Hutton A, Malouf D, Lee D, Cozzi P, MacLean F, Thompson J.BMC Urol. 2021 Apr 22;21(1):67. doi: 10.1186/s12894-021-00832-5.PMID: 3388812
Piscaglia F, Nolsøe C, Dietrich CF, Cosgrove DO, Gilja OH, Bachmann Nielsen M, Albrecht T, Barozzi L, Bertolotto M, Catalano O, Claudon M, Clevert DA, Correas JM, D'Onofrio M, Drudi FM, Eyding J, Giovannini M, Hocke M, Ignee A, Jung EM, Klauser AS, Lassau N, Leen E, Mathis G, Saftoiu A, Seidel G, Sidhu PS, terHaar G, Timmerman D, Weskott HP. The EFSUMB Guidelines and Recommendations on the Clinical Practice of Contrast Enhanced Ultrasound (CEUS): update 2011 on non-hepatic applications. Ultraschall Med 2012; 33: 33-59 [PMID: 21874631 DOI: 10.1055/s-0031-1281676] 7
Diagnostic Performance of Contrast-Enhanced Ultrasound in the Evaluation of Small Renal Masses: A Systematic Review and Meta-Analysis.
Tufano A, Antonelli L, Di Pierro GB, Flammia RS, Minelli R, Anceschi U, Leonardo C, Franco G, Drudi FM, Cantisani V.Diagnostics (Basel). 2022 Sep 25;12(10):2310. doi: 10.3390/diagnostics12102310.PMID: 36291999
Diagnostic Accuracy of MRI for Solid Renal Masses: A Systematic Review and Meta-analysis.
Frank RA, Dawit H, Bossuyt PMM, Leeflang M, Flood TA, Breau RH, McInnes MDF, Schieda N.J Magn Reson Imaging. 2023 Apr;57(4):1172-1184. doi: 10.1002/jmri.28397. Epub 2022 Aug 20.PMID: 36054467
Needle tract seeding after percutaneous cryoablation of small renal masses; a case series and literature review.
Rizzo M, Cabas P, Pavan N, Umari P, Verzotti E, Boltri M, Stacul F, Bertolotto M, Liguori G, Trombetta C.Scand J Urol. 2020 Apr;54(2):122-127. doi: 10.1080/21681805.2020.1736149. Epub 2020 Mar 10.PMID: 32153242
The R.E.N.A.L. nephrometry score: a comprehensive standardized system for quantitating renal tumor size, location and depth.
Kutikov A, Uzzo RG.J Urol. 2009 Sep;182(3):844-53. doi: 10.1016/j.juro.2009.05.035. Epub 2009 Jul 17.PMID: 1961623
Active Surveillance of Small Renal Masses: A Review on the Role of Imaging With a Focus on Growth Rate.
Nayyar M, Cheng P, Desai B, Cen S, Desai M, Gill I, Duddalwar V.J Comput Assist Tomogr. 2016 Jul-Aug;40(4):517-23. doi: 10.1097/RCT.0000000000000407.PMID: 2733192
Recent advances in imaging techniques of renal masses.
Aggarwal A, Das CJ, Sharma S.World J Radiol. 2022 Jun 28;14(6):137-150. doi: 10.4329/wjr.v14.i6.137.PMID: 35978979
Reviewer 2 Report
Comments and Suggestions for Authors
The authors should be congratulated for their work. The need to improve renal cell cancer (RCC) imaging is due to the aggressiveness of this cancer which nowadays does not be screened into screening programs. The majority of the diagnosis indeed are incidental. The review had a solid basis to be comprehensive but lacked clinical vision. Specifically, the current manuscript sounds like a sterile list of stuff. The importance of the current paper should be acknowledged by the authors first and then by the urological community which could gain only benefits.
First, the imaging techniques addressed renal cell cancer. Any information for the histological subtypes? Such as chromophobe, collecting duct carcinoma, or sarcomatoid variant (PMID= 23434943, 36456452, 38773038). This aspect should be discussed due to the higher aggressiveness of the sarcomatoid variant and collecting duct carcinoma compared to RCC. May the radiomics help the clinicians and surgeons in some way?
Second, one of the major concerns of RCC imaging is the lack of standardization for detecting recurrence of the disease after Partial Nephrectomy. Does the radiomics help the clinicians after a PN with positive surgical margins? (PMID 25450033).
Third, a PRISMA is needed even if the review has not a systematic design. It would be useful to understand the quantity of the data (for sure not a paucity) on which the Authors relied on.
Reviewer 3 Report
Comments and Suggestions for Authors
Review paper “Imaging in Renal Cell Carcinoma Detection”
Dixon Woon et al., provide a systematic review about imaging in Renal Cell Carcinoma (RCC). The main object of this paper is to review the literature about RCC imaging methods. Several known imaging techniques, including recent advances in nuclear medicine, are reviewed. However, other imaging methods such as Dual Energy CT and Photon Counting CT should be covered more comprehensively, as well as the paragraph of radiomics and artificial intelligence which lacks summary table like the other paragraphs.
Although the literature analysis is consistent, same papers are not considered in the analysis such “Update on Renal Cell Carcinoma Diagnosis with Novel Imaging Approaches” of Marie-France Bellin et al. which provides a very useful update on recently new approaches on RCC imaging and should absolutely be reported.
Overall English is good, taking into account some minor grammatical errors.
The methods are clearly illustrated and efficient, as well as the conclusions.
Comments on the Quality of English LanguageOverall English is good, taking into account some minor grammatical errors.
Round 2
Reviewer 1 Report
Comments and Suggestions for Authors
No addtional comments.
In my opinion improved manuscript draft, and more contemporary information, probably relevant for the clinical practice of the less experienced abdominal radiologist.
Reviewer 2 Report
Comments and Suggestions for Authors
The authors addressed properly my comments.
Reviewer 3 Report
Comments and Suggestions for Authors
The requested changes have been excellently made, making the article more complete and detailed, especially taking into account the recent updates with new techniques.
Minimal English language errors have been resolved.